# The immune protection induced by a serine protease from the *Trichinella spiralis* adult against *Trichinella spiralis* infection in pigs

**Daoxiu Xu**[1,2]☯, **Xue Bai**[1]☯, **Jing Xu**[3]☯, **Xuelin Wang**[1], **Zijian Dong**[1], **Wenjie Shi**[1], **Fengyan Xu**[1], **Yanfeng Li**[1], **Mingyuan Liu**[1,4]*, **Xiaolei Liu**[1]*

**1** Key Laboratory of Zoonosis Research, Ministry of Education, Institute of Zoonosis, College of Veterinary Medicine, Jilin University, Changchun, China, **2** Department of Human Parasitology, School of Basic Medicine, Hubei University of Medicine, Shiyan, China, **3** College of Animal Science and Technology, Inner Mongolia University for Nationalities, Tongliao, China, **4** Jiangsu Co-innovation Center for Prevention and Control of Important Animal Infectious Diseases and Zoonoses, Yangzhou, China

☯ These authors contributed equally to this work.
* liumy36@163.com (ML); liuxlei@163.com (XL)

**Data Availability Statement:** All relevant data are within the manuscript.

**Funding:** MYL was supported by the National Key Research and Development Program of China

## Abstract

Trichinellosis is a major foodborne parasitosis caused by *Trichinella spiralis*. In the present study, a serine protease gene from an adult *T. spiralis* (Ts-Adsp) cDNA library was cloned, expressed in *Escherichia coli* and purified by Ni-affinity chromatography. Previous studies of our laboratory have found that mice vaccinated with recombinant Ts-Adsp protein (rTs-Adsp) exhibited partial protection against *T. spiralis* infection. In this study, the protective effect of rTs-Adsp against *T. spiralis* infection in pigs was further explored. The cell-mediated and humoral immune responses induced by rTs-Adsp were measured, including the dynamic trends of specific antibody levels (IgG, IgG1, IgG2a and IgM), as well as the levels of cytokines (IFN-γ, IL-2, IL-4, and IL-10) in the serum. Moreover, the changes in T lymphocytes, B lymphocytes, and neutrophils were measured to evaluate cellular immune responses in pigs vaccinated with rTs-Adsp. The results indicated that a Th1-Th2 mixed immune response with Th1 predominant was induced by rTs-Adsp after vaccination. Flow cytometric analysis showed that the proportions of CD4[+] T cells, B cells, and neutrophils in the immunized groups were significantly increased. Furthermore, pigs vaccinated with rTs-Adsp exhibited a 50.9% reduction in the muscle larvae burden, compare with pigs from the PBS group five weeks after challenged. Our results suggested that rTs-Adsp elicited partial protection and it could be a potential target molecule for preventing and controlling *Trichinella* transmission from pigs to human.

## Author summary

Trichinellosis is a global foodborne parasitic disease caused by consuming raw or poorly cooked meat. The porcine products are the most common source. Therefore, it will have a great significance for public health security and human health to prevent and control the trichinellosis. We previously found that mice vaccinated with recombinant Adsp protein

(2018YFC1602500); National Natural Science Foundation of China (NSFC 31872467, 31520103916); Jilin Provincial Science and Technology Development Project (20180520042JH); and Program for JLU Science and Technology Innovative Research Team (2017TD-32). The funders had no role in study design, data collection and analysis, decision to publish, or preparation of the manuscript.

**Competing interests:** The authors have declared that no competing interests exist.

(rTs-Adsp) exhibited partial protection against *T. spiralis* infection. In this study, the protective effect of rTs-Adsp against challenge infections with *T. spiralis* in pigs was further explored. We found that rTs-Adsp elicited partial protection and it could be an important target molecule for preventing and controlling *T. spiralis* transmission from pigs to human.

## Introduction

*Trichinella spiralis* (*T. spiralis*) is a foodborne parasite that can infect a wide range of animals, such as mammals, birds and reptiles [1]. Trichinellosis caused by *T. spiralis* is a public health hazard and can affect food safety, especially pork-related products [1]. The source of human infection is mainly digestion of raw or poorly cooked meat, and porcine products are the most common source [2–4]. Therefore, the important measure to control trichinellosis should prevent the transmission from pigs to human [5]. The development of vaccines against *T. spiralis* infection in pigs might be a promising method of parasite control. However, so far, most studies on *Trichinella* vaccines have been performed in mouse models, and very few anti-*Trichinella* infection studies have been performed on pigs [6]. The exploitation of vaccines against *T. spiralis* infection in pigs is an important measure by which to block infection from pigs to humans [6, 7].

Proteases are a type of enzymes that are widely distributed in eukaryotes, prokaryotes, and viruses [8]. Research has shown that serine proteases participate in many different events in the life cycle of a parasite [8]. Serine proteases from parasites are thought to be key factors in the process of establishing infection. Moreover, serine proteases are important enzymes that exist in excretory-secretory (ES) production of *T. spiralis*. Many studies have shown that proteases exhibited immune protection effects against *T. spiralis* infection in mice [9–13]. Among these proteases, serine proteases play a crucial role in *T. spiralis* invasion of host cells and are involved in the processes of immune evasion [14]. There is much evidence showing that serine proteases in parasite ES products perform multiple functions and involve the processes of parasite nutrition and immune evasion [15–17]. In addition, serine proteases have been proven to be involved in parasite survival, and these proteases have been considered as candidate antigens for vaccines against parasite infection [18–20].

The entire life cycle of *T. spiralis* is divided into three major developmental stages: adult worms (AD), newborn larvae (NBL), and muscle larvae (ML) [21]. Serine protease-like antigens have been identified in adult stages of *T. spiralis*. Adult stage is a major stage for the reproduction of *T. spiralis*. If growth from ML to AD is interrupted, the quantity of NBL will be reduced, which will hamper or reduce the production of ML in the muscle [22]. It was reported that a serine protease (Ts-Adsp) was screened from *T. spiralis* adult and that mice vaccination with recombinant Adsp protein (rTs-Adsp) elicited a 46.5% reduction in ML [9]. Sun et al. screened a serine protease (TsSP) from ES products of *T. spiralis* and found that mice vaccinated with recombinant TsSP protein (rTsSP) exhibited 52.70% and 52.10% reduction in AD and ML, respectively [7]. Moreover, a previous study of our laboratory reported that serine protease from *T. spiralis* adult might be participated in the process of capsule formation and protect the NBL in the circulatory system of the host [23]. Therefore, serine proteases from *T. spiralis* adults have become a promising vaccine targets.

In recent years, a number of vaccines against *T. spiralis* have been developed to prevent transmission from animals to humans. Most potential vaccine candidates were selected from recombinant proteins and ES products. A previous study found that serine protease from *T.*

*spiralis* adults exhibited partial protection effects against challenging with *T. spiralis* larvae [9]. In this study, the serine protease gene of *T. spiralis* was identified from a cDNA library of *T. spiralis* adult worms (Ts-Adsp) and purified by Ni-affinity chromatography. To further explore the protective effect of the rTs-Adsp protein, pigs were immunized with the rTs-Adsp protein to evaluate its immune protection effects against *T. spiralis* infection. Studies have found that high effective vaccine efficacy against trichinellosis are associated with high levels of humoral and cellular immune response in mice [7, 24, 25]. The antibodies can kill the ML, reduce the larval infectivity, and obstruct larval development through an ADCC mechanism [26]. More-over, the expulsion of *T. spiralis* adults is mainly regulated by CD4$^+$/Th2 cytokines and depends on the production of IL-4 and IL-13, since inhibition of these cytokines extends parasite survival [27]. In this study, we have provided the possible immune mechanism that occurs after immunization with rTs-Adsp protein in pigs.

## Materials and methods

### Ethics statement

All animals were raised carefully in accordance with the Animal Ethics Procedures and Guidelines of the People's Republic of China. The protocol of the animal experiments was reviewed and approved by the Institutional Animal Care and Use Committee of Jilin University (protocol # 20170318).

### Parasite and animals

In this study, *T. spiralis* (ISS534) was obtained from Wistar rats by pepsin–HCl digestion. Eighteen two-month-old large white pigs were obtained from a standardized pig farm (Jilin Province, China). All pigs had not been vaccinated with a *T. spiralis* vaccine. Moreover, all pigs were fed an antibiotic-free energy diet. Before the experiment, blood from all pigs were collected to analyze the blood routine examination. The fecal sample were collected to test other parasite eggs by flotation and sedimentation method. All pigs were kept under standard pig houses in our laboratory and underwent a week of health observation.

### Preparation of plasmid pET28a-Ts-Adsp

The coding sequence of the Ts-Adsp gene (GenBank EU263332.1) was amplified from a cDNA library of *T. spiralis* adult worms by PCR (forward primer 5′-<u>GAATCC</u>GAATTAT-GAATGTG-3′, containing the *EcoRI* restriction site, and reverse primer 5′-<u>CTCGAG</u>ACG-GAAAAAAGT-3′, containing the *XhoI* restriction site). The PCR product was cloned into prokaryotic expression vector pET28a and transformed into *Escherichia coli* (*E. coli*) BL21 (DE3) cells.

### Preparation and identification of recombinant Ts-Adsp (rTs-Adsp)

The complete sequence of the Ts-Adsp gene was cloned into the pET28a expression vector using T4 DNA ligase. Then, the plasmid pET28a-Ts-Adsp was transformed into *E. coli* BL21 (DE3) cells. The expression of rTs-Adsp was induced with 1mM IPTG (isopro-pyl-β-D-thioga-lactopyranoside) for 6 h at 20˚C. Then, purification of the rTs-Adsp was performed as previously described [28]. Finally, the rTs-Adsp protein was identified by 12% SDS–PAGE and western blot analysis, as previously described [13]. Briefly, the rTs-Adsp protein was elec-tro-transferred to a nitrocellulose membrane. After blocking with 5% skimmed milk, the membrane was incubated with serum from pig infected with 20000 *T. spiralis* ML at a 1:200 dilution in TBST overnight. The serum from *Trichinella*–negative pig served as a negative

control. Then the membrane was incubated with HRP-conjugated goat anti-swine IgG (Bio-Rad) at a 1:4000 dilution in TBST for 2 h. Finally, the membrane was incubated with the enhanced chemiluminescence (ECL) reagents and subsequently photographed on BioMax film. Contaminating endotoxin was removed from the rTs-Adsp protein by a ToxOut High-Capacity Endotoxin Removal Kit (Biovision, USA) according to the manufacturer's instructions. Briefly, at least four endotoxin standard solutions were prepared to generate a standard curve. The absorbance of each reaction was read at 545 nm. Endotoxin concentrations of samples were calculated from the standard curve. The residual endotoxin in the rTs-Adsp protein was approximately 0.1431 EU/mL.

## Immunization procedure and challenge

Montanide IMS 1313 N VG PR (IMS1313) was obtained from SEPPIC Corporation (Paris, France). Eighteen piglets were randomly divided into 3 groups, namely the PBS group, IMS1313 group (with a 1:1 dilution in PBS) and rTs-Adsp protein group (with a 1:1 dilution in IMS1313). In total, six pigs in each group were immunized twice at a four-week interval. In the rTs-Adsp protein group, each pig was immunized with 1 mg rTs-Adsp (1 mg/mL). In the group of PBS and IMS1313, each pig was immunized with 2 mL PBS, IMS1313, respectively. Three weeks after the last immunization, each group of pigs was inoculated with 5000 *T. spiralis* muscle larvae by oral administration.

## Specific antibody response

Blood from pigs in each group was collected to evaluate the changes in specific antibodies (IgG, IgG1, IgG2a and IgM) by an indirect enzyme-linked immunosorbent assay. Briefly, microtiter plates were coated with 100 μL purified protein at 4°C overnight. The titers of specific IgG and IgG subtypes were detected as described previously [28]. For IgM detection, the titers of specific IgM were detected as described previously [28].

## Cytokine assays

Serum samples of pigs were obtained two weeks after the last immunization. Cytokines (IFN-γ, IL-2, IL-4, and IL-10) in the serum samples were detected using ELISA kits (R&D Systems, Minneapolis, MN, USA) according to the manufacturer's instructions. The levels of cytokines in the serum were extrapolated according to the standard curve provided by the ELISA kit.

## Flow cytometry

Blood from six pigs in each group was collected for flow cytometry analysis. The relative ratios of T cells, B cells, and neutrophils were evaluated by flow cytometry. Briefly, the cells were obtained from blood lysed by red blood cell lysis buffer (Solarbio, Beijing, China). For antibody labeling of T cells, the cells were stained with 2 μL mouse anti-porcine CD3e-FITC (SouthernBiotech, Birmingham, AL, USA), 2 μL mouse anti-porcine CD4-PE (SouthernBiotech) and 2 μL mouse anti-porcine CD8a-PE-Cy5 antibodies (Abcam, Cambridge, MA, USA). For antibody labeling of B cells, the cells were stained with 6 μL mouse anti-porcine CD3e-FITC (SouthernBiotech) and 2 μL mouse anti-porcine CD21-PE antibodies (Abcam). For antibody labeling of neutrophils, the cells were stained with 10 μL mouse anti-porcine CD14-PE (Abcam), 5 μL mouse anti-porcine SWC8 (Abcam) and 5 μL goat anti-mouse IgM-FITC antibodies (SouthernBiotech). Then, the cells and antibodies were incubated on ice for 30 min in the dark. After washing twice with PBS, the stained cells were resuspended in 300 μL of PBS.

Finally, the stained cells were measured by a flow cytometer (BD Biosciences). The percentages of T cells, B cells, and neut.rophils were analyzed using FlowJo.

### Evaluation of larval burden

Six pigs in each group were sacrificed five weeks after challenge. Six different types of muscles were obtained from each pig, respectively, including the diaphragm, tongue, gastrocnemius, trapezius, gluteus maximus, and flexor tendon. Fifty grams of muscle of each type were collected to evaluate immune protection. Finally, the number of muscle larvae per gram of muscle (LPG) and the worm reduction rate were evaluated by comparison with those of the PBS group.

### Statistical analysis

GraphPad Prism 8.0 software was used to perform statistical analysis. Two experimental groups were compared using Student's t-test for nonparametric data. Three groups were compared using ANOVA with Dunnett's multiple comparison test as indicated. Normality (Shapiro-Wilk test) and homogeneity of variance (Levene's test) were performed in each case. The data are displayed as the means ± standard deviations (SD). The significance of the differences between the groups was expressed as $^{*}$p$<$ 0.05, $^{**}$p$<$ 0.01, and $^{***}$p$<$ 0.001.

## Results

### Analysis of rTs-Adsp protein

SDS-PAGE results indicated that the molecular mass of the rTs-Adsp protein was approximately 47.5 kDa, and the purified protein was visualized as a single band (Fig 1A). Western blotting results indicated that the rTs-Adsp protein was recognized by the serum of pig infected with *T. spiralis* for 60 days (Fig 1B). The results suggested that the rTs-Adsp protein was purified successfully and had good reactogenicity.

### Immune responses in vaccinated pigs

To determine the antibody response to rTs-Adsp in vaccinated pigs, serum was collected before and after vaccination. In IgG and IgM detection, compared with those of pigs immunized with PBS, the levels of specific IgG and IgM in pigs immunized with the rTs-Adsp protein were significantly elevated (Fig 2A and 2B). IgG subtypes detection results showed that the levels of IgG subtype in pigs immunized with the rTs-Adsp protein were very significantly elevated (Fig 2D and 2E). Moreover, the titers of IgG2a were higher than those of IgG1 (Fig 2C), suggesting that a Th1-dominant Th1/Th2 mixed immune response was induced by the rTs-Adsp protein.

### Evaluation of cytokine production

To evaluate the cell-mediated immune response of mice vaccinated with rTs-Adsp, the concentration of cytokines (IL-2, IFN-γ, IL-4, and IL-10) associated with Th1and Th2 were examined by ELISA. The results showed that the titers of cytokines in pigs immunized with rTs-Adsp protein were significantly elevated compared with those of the PBS group; however, there was no significant change between the PBS group and the adjuvant group (Fig 3). The results confirmed that a Th1 and Th2 mixed immune response was induced after immunization with rTs-Adsp.

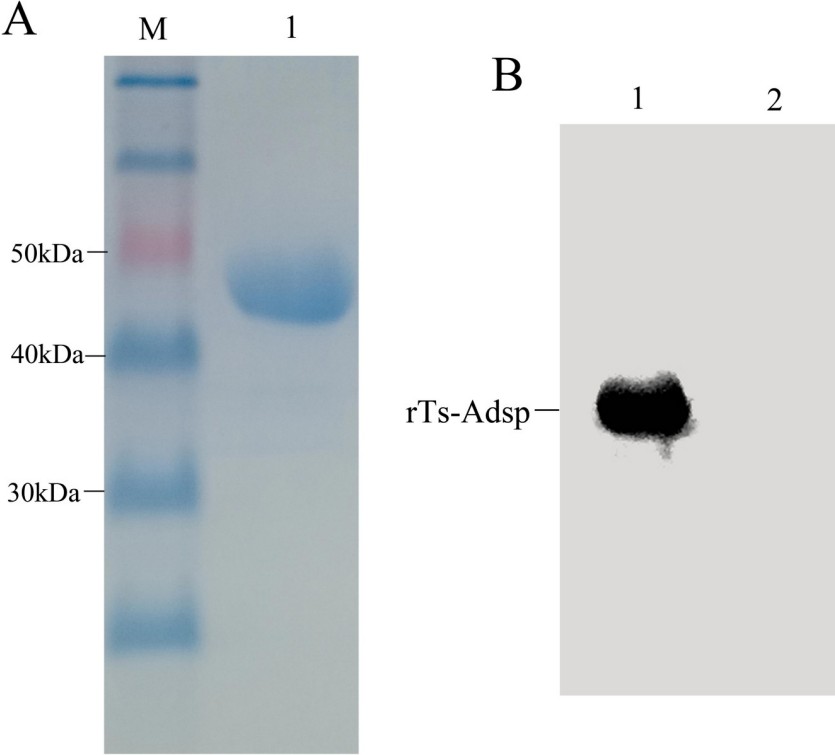

**Fig 1. Purification and identification of recombinant rTs-Adsp by SDS-PAGE and western blot.** (A) Purified rTs-Adsp was analyzed by SDS-PAGE. Lane M: protein molecular weight marker, Lane 1: purified rTs-Adsp by Ni-affinity chromatograph. (B) The antigenicity of rTs-Adsp was identified by western blot. Lane 1: rTs-Adsp incubated with serum from pig infected with *T. spiralis* at 60 dpi, Lane 2: rTs-Adsp incubated with serum from *Trichinella*–negative pig.

## Evaluation of the changes in T lymphocytes

Changes in T cell subsets (CD4$^+$ T and CD8$^+$ T) are the indicators of the immune function status. It is well known that CD4$^+$ T cells play a key role in regulating immune response. Two weeks after the last vaccination, compared with that of pigs immunized with PBS, the proportion of CD4$^+$ T cells in pigs immunized with the rTs-Adsp protein was elevated, and the proportion of CD8$^+$ T cells did not change significantly (Fig 4). The results suggested that the cellular immune response in pigs was induced after vaccination with rTs-Adsp.

## Evaluation of the change in B lymphocytes

The humoral immune response ability in a host is related to the changes of B lymphocytes. The proliferation of B cells is required for antibody production, which plays an important role in anti-*Trichinella* immunity. Two weeks after the last vaccination, compared with that of pigs immunized with PBS, the proportion of B cells in pigs immunized with the rTs-Adsp protein was significantly elevated (Fig 5). The results suggested that the humoral immune response in pigs was enhanced after immunization with rTs-Adsp.

## Analysis of the changes in neutrophils

It is well known that neutrophils play crucial roles in nonspecific immune defense by phagocytizing foreign bodies. Two weeks after the last vaccination, the percentage of neutrophils in the

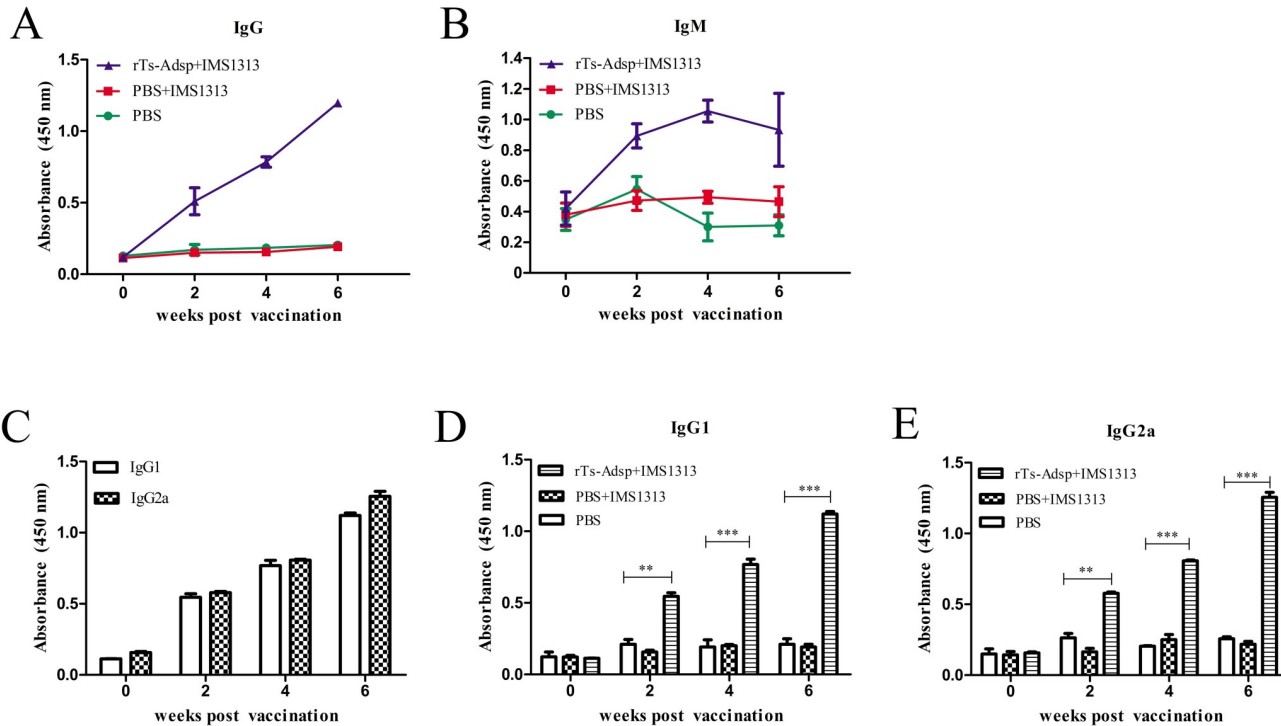

**Fig 2. Immune responses from the vaccinated pigs.** (A) The levels of IgG against rTs-Adsp were measured by ELISA. (B) The levels of IgM in the sera of immunized pigs were measured by ELISA. (C) IgG subclass responses to the rTs-Adsp were measured by ELISA. (D) The IgG1 subclass responses against rTs-Adsp were evaluated at different time points. (E) The IgG2a subclass responses against rTs-Adsp were evaluated at different time points. The values shown for each group are the mean ± SD of the antibody levels. Significant differences were as follows: $^*$p< 0.05; $^{**}$p< 0.01; $^{***}$p< 0.001.

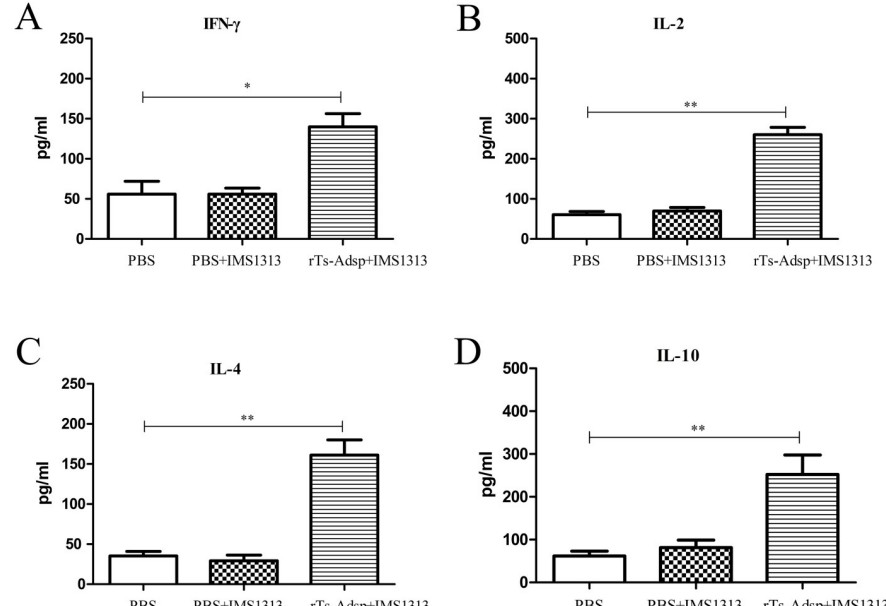

**Fig 3. Cytokine production from sera were evaluated by ELISA.** The levels of (A) IFN-γ (B) IL-2 (C) IL-4 (D) IL-10 are presented as the mean ± SD (n = 6). Asterisks indicate that the production of cytokines from immunized group are significantly different from ($^*$p<0.05, $^{**}$p<0.01, $^{***}$p< 0.001) that of PBS control group.

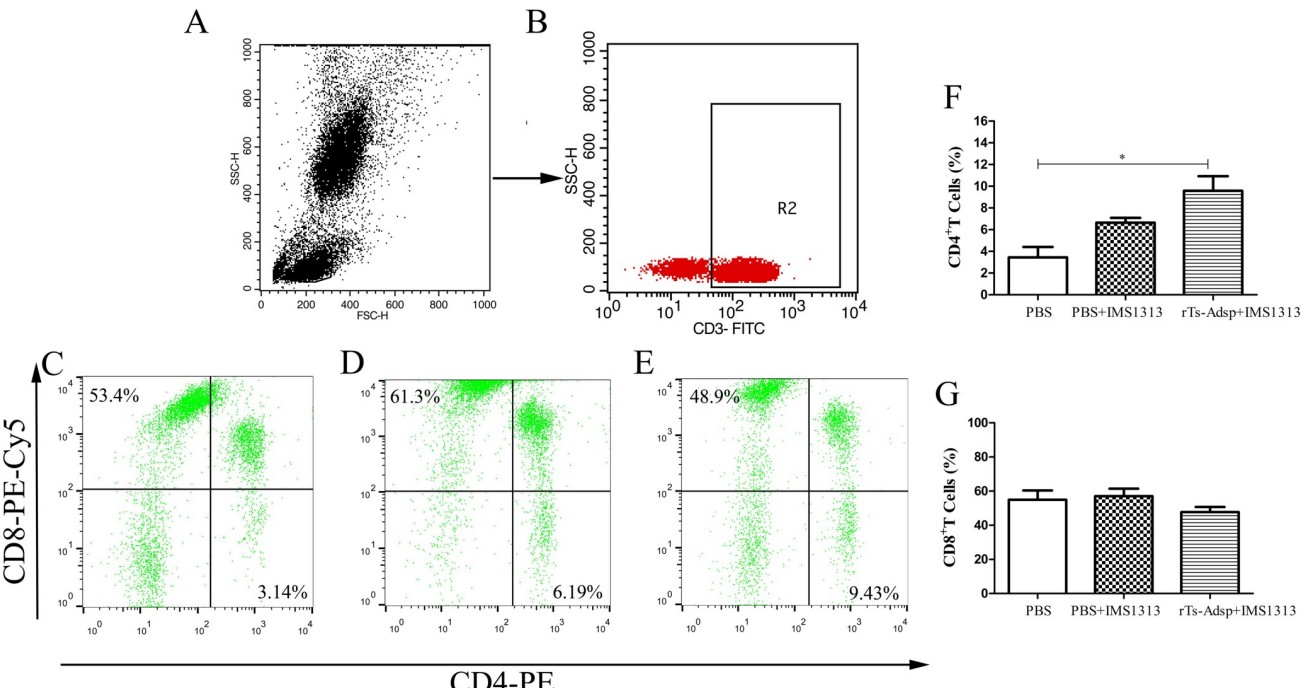

**Fig 4. Flow cytometry analysis of T lymphocytes in peripheral blood.** The gating strategy are shown in (A) and (B). The percentage change of CD4+ and CD8+ T cells in peripheral blood lymphocytes from (C) PBS-control group (D) IMS1313 adjuvant group (E) rTs-Adsp group. Statistical analysis of (F) the percentage of CD4+ T lymphocytes (G) the percentage of CD8+ T lymphocytes. The values are shown as mean ± SD. Significant differences are as follows: *p<0.05, **p<0.01, ***p<0.001.

peripheral blood was analyzed using flow cytometry. The results showed that the proportion of neutrophils in the immunized group was significantly elevated compared with the control group (Fig 6). The results indicated that the immune defense ability of pigs was enhanced after vaccination with rTs-Adsp.

## Immune protection

The immune protection of rTs-Adsp against *T. spirails* infection was evaluated in immunized pigs. Compared with pigs in the PBS-immunized group, pigs in the rTs-Adsp-immunized group showed a 50.9% reduction in the muscle larvae burden (Fig 7). The muscle larvae burden in the immunized group was significantly lower than that of the PBS group (p< 0.05). However, compared with the PBS group, the muscle larvae burden in the adjuvant group did not change significantly (Fig 7). The results demonstrated that pigs immunized with rTs-Adsp exhibited partial protection against *T. spiralis* infection.

## Discussion

Over the past decades, many efforts have been made to study *T. spiralis* vaccines, which have contributed to the development of novel vaccines. A number of vaccine strategies have been evaluated, including recombinant protein vaccines, DNA vaccines, and synthesized epitope vaccines [5]. Among them, protein vaccines are a major strategy used to control *T. spiralis* infection. Moreover, ES products, proteases, whole worms, and surface proteins are mainly selected as candidate immunogens. Serine proteases have been identified in ES production and are thought to be involved in the process of *T. spiralis* moluting and invasion of host cells [14]. The host will produce specific antibodies and cellular immune responses after

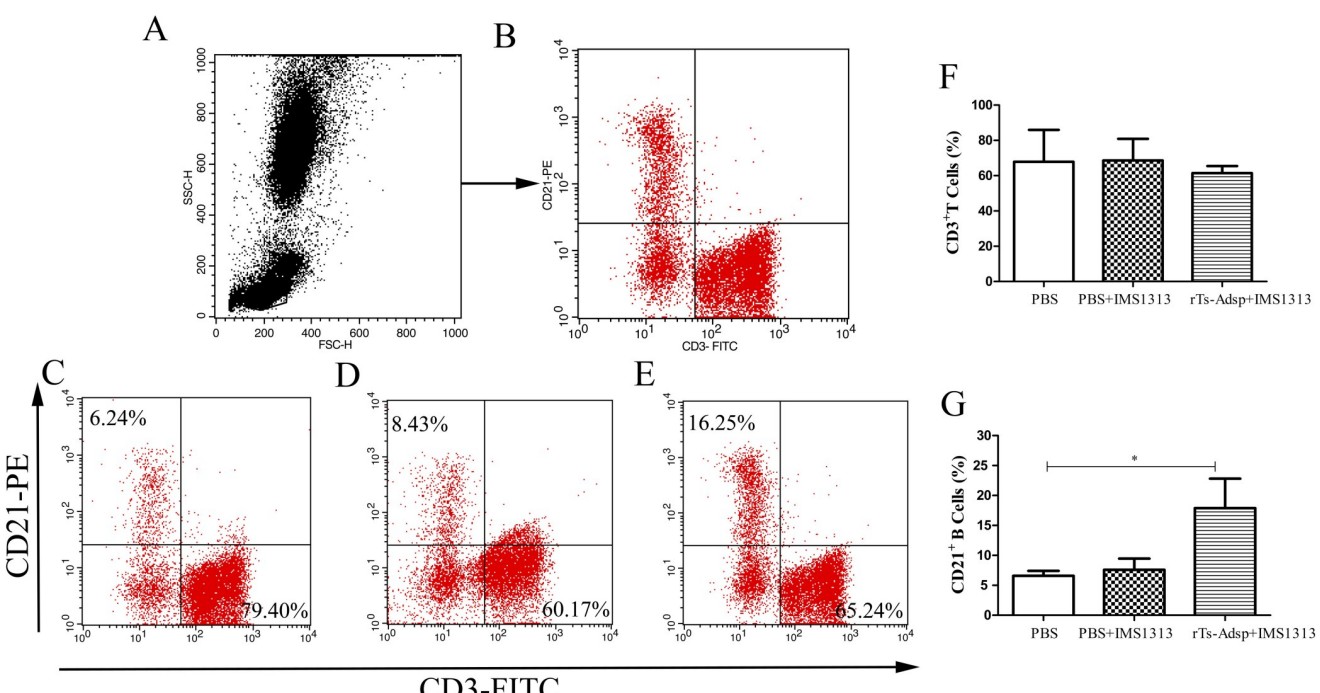

**Fig 5. Flow cytometry analysis of B lymphocytes in peripheral blood.** The gating strategy are shown in (A) and (B). The populations change of B lymphocytes in peripheral blood from (C) PBS-control group (D) IMS1313 adjuvant group (E) rTs-Adsp group. Statistical analysis of (F) the percentage of CD3$^+$ T lymphocytes (G) the percentage of CD21$^+$ B lymphocytes. The values are shown as mean ± SD. Significant differences are as follows: $^*$p<0.05, $^{**}$p<0.01, $^{***}$p<0.001.

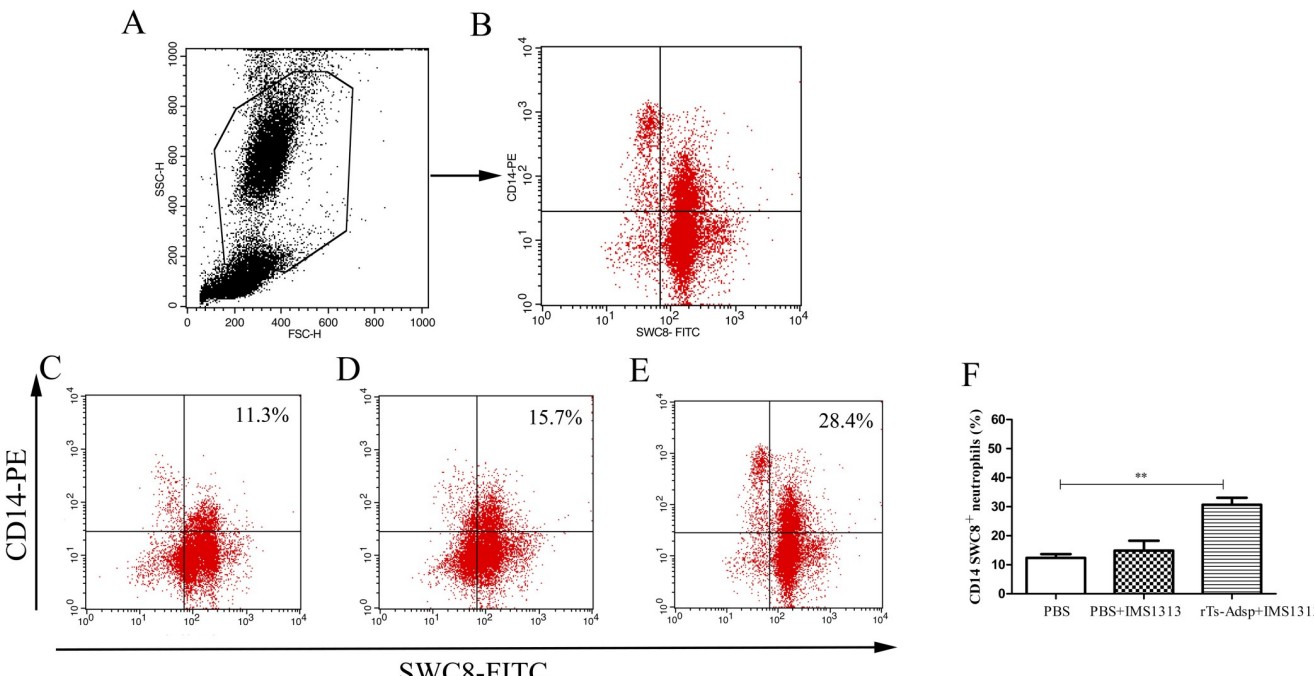

**Fig 6. Flow cytometry analysis of neutrophils in peripheral blood.** The gating strategy are shown in (A) and (B). The percent change of neutrophils in peripheral blood from (C) PBS-control group (D) IMS1313 adjuvant group (E) rTs-Adsp group. Statistical analysis of (F) the percentage of neutrophils. The values are shown as mean ± SD. Significant differences are as follows: $^*$p<0.05, $^{**}$p<0.01, $^{***}$p<0.001.

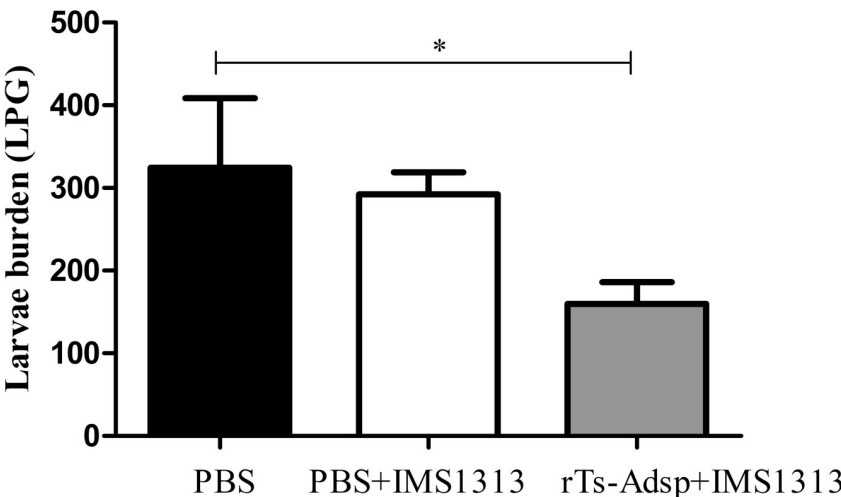

**Fig 7. Protective immunity of rTs-Adsp-vaccinated pigs after being challenged with 5000 *Trichinella spiralis* larvae.** The results are shown as the mean ± SD (n = 6). Asterisks indicate that muscle larvae burden of immunized group is significantly different from (*p<0.05) that of PBS control group.

vaccination with serine protease-like antigens. Inhibition of the activities of proteases by specific antibodies may have an adverse impact on parasite survival [7, 10, 29, 30]. Sun et al. reported that antibodies induced by a serine protease-like antigen can inhibit larval invasion of the host enterocytes [7]. Moreover, the antibodies can kill the ML, reduce the larval infectivity, and obstruct larval development through an ADCC mechanism [26]. A previous study of our laboratory found that a serine protease from *T. spiralis* adults exhibited partial protection against challenge with *T. spiralis* larvae [9]. Therefore, serine proteases could be a potential target molecule for preventing and controlling *T. spiralis* infection. In this study, the serine protease from *T. spiralis* adults was expressed and identified to investigate its protective immunity effect against *T. spiralis* in pigs.

It is well known that the protective effect of most vaccines is related to the antibody responses [31, 32]. Antigens are often screened to develop anti-parasite vaccines based on the degree of antibody responses; thus, vaccines that show high-level protection are usually related to effective antibody responses [33, 34]. Previous studies have found that mice immunized with serine protease-like antigens exhibited high-level cell-mediated and humoral immune responses, eliciting partial protection against challenge with *T. spiralis* larvae [9, 13, 35]. Most studies have demonstrated that host infection with *T. spiralis* induced a Th2-based immune response, which is probably related to the immunosuppressive effect of *T. spiralis* on the host [36–38]. It is well known that the IgG1 isotype represents a Th2 immune response, whereas the IgG2a isotype represents a Th1 immune response [39]. In this study, the levels of antibody responses (IgG, IgG1, IgG2a, and IgM) induced by rTs-Adsp in pigs were analyzed to investigate the protective immunity effect of rTs-Adsp. The results showed that the levels of antibodies in the immunized group were significantly elevated compared to those in the control group. The results showed that a mixed IgG1 and IgG2a antibody response, with IgG2a antibody response being predominant, was induced by the rTs-Adsp protein. Although antibody response is necessary for protective immunity, cell-mediated immunity is also important in eliminating parasites [40]. Furthermore, the titers of cytokines (IL-2, IFN-γ, IL-4, and IL-10) in the serum of pigs were measured. The results showed that the levels of cytokines in the immunized group were significantly elevated compared to the control group. A study found that the expulsion of *T. spiralis* adults is mainly regulated by CD4$^+$/Th2 cytokines and depends

on the production of IL-4 and IL-13, since inhibition of these cytokines extends parasite survival [27]. IL-10 plays a major role in regulating intestinal mast cell responses and is important in protecting against *T. spiralis* adults [41]. Furthermore, IFN-γ is involved in killing the newborn larvae of *T. spiralis* by activating macrophages and enhancing the cytotoxic killing of eosinophils and granulocytes [40]. IL-2 is crucially involved in resisting acute acquired infection [42]. The above results demonstrated that rTs-Adsp induced humoral and cell-mediated immune responses.

A study found that *T. spiralis* has an immunosuppressive effect on the natural immune system, and the inhibitory effect is most significant for the adult stage and newborn larval migration stage of *T. spiralis* [43]. Eliminating the immunosuppression caused by *T. spiralis* infection plays a major role in killing the parasites. Neutrophils are crucially components of the innate immune response and can regulate adaptive immune responses [44–46]. It was reported that neutrophils can kill worms, depending on antibody dependent cell-mediated cytotoxicity (ADCC) [47]. This effect has been confirmed through in vitro experiments and the development of a *T. spiralis* vaccine [48, 49]. Moreover, neutrophils have the function of promoting inflammation and fighting pathogen infection [38, 44]. A previous study of our laboratory found that the level of neutrophils was increased after *T. spiralis* infection [38]. In this study, compared to that of the PBS group, the proportion of neutrophils in the immunized group was increased. The change may contribute to expelling worms and eliminating the immunosuppression caused by *T. spiralis* infection. Furthermore, the results of T and B lymphocytes indicated that the numbers of CD4$^+$ T lymphocytes and B lymphocytes in the rTs-Adsp group were significantly elevated compared to those of the PBS group. The numbers of CD8$^+$ T lymphocytes was decreased, but not statistically significant compared to those of the PBS group. It was reported that the expulsion of *T. spiralis* adults is dependent on CD4$^+$ T cells [50]. The resistance of infective muscle larvae is partially dependent on related cytokines regulated by CD4$^+$ T cells [36, 51]. Study found that CD8$^+$ T cells play no significant role in worm expulsion, but that CD4$^+$ T cells may make a significant contribution [52]. Meanwhile, the ratio of CD4$^+$/CD8$^+$ T cells may reflect the ratio of helper T cells to suppressor T cells, and the immune status of the host may be determined by the balance of these cells [53]. In the present study, the ratio of CD4$^+$/CD8$^+$ T cells were elevated. The result indicated that the cellular immune response in pigs was induced after vaccination with rTs-Adsp. B cells can elicit different functions; they can secrete immunoglobulin, produce cytokines, and induce Treg cell production [54, 55]. It is well known that antibodies produced by B cells play a key role during *T. spiralis* infection. One study reported that IgG antibodies produced by B cells can kill NBL by induction of eosinophils and ADCC [53]. The above results suggested that rTs-Adsp induced a cell-mediated immune response, which is likely to play a major role in protecting the host against *T. spiralis* infection.

Until now, most studies on *Trichinella* vaccines have been performed in mouse models, and very few anti-*Trichinella* infection studies have been performed on pigs. In our previous study, pig vaccinated with DNase II-7/FCA exhibited a 45.7% reduction in the muscle larvae burden [28]. However, due to the adverse reaction of Freund's adjuvant to experimental animals, its application is gradually limited. Montanide IMS 1313 N VGPR adjuvant used in the present study is comparatively nontoxic and has been applied to animal vaccine research. Previous study found that compared to FCA/FIA-formulated vaccines, an IMS1313/rTs-serpin vaccine displayed higher levels of immune response and protective efficacy [56]. Obviously, the protective effect is crucially associated with the reduction in the muscle larvae burden. A previous study of our laboratory found that mice vaccinated with rTs-Adsp exhibited a 46.5% reduction in muscle larvae [9]. Moreover, mice immunized with pcDNA3.1(+)-Ts-Adsp showed a 50.2% ML reduction, and a combination of pcDNA3.1(+)-Ts-Adsp and rTs-Adsp

elicited a 69.5% ML reduction [35]. In the present study, the protective immunity effect of rTs-Adsp against *T. spiralis* infection in pigs was further explored. The results showed that rTs-Adsp elicited a 50.9% ML reduction in pigs.

In conclusion, rTs-Adsp induced humoral and cell-mediated immune responses in vaccinated pigs. Furthermore, pigs immunized with rTs-Adsp exhibited partial protection against challenge with *T. spiralis* larvae. Our results indicated that rTs-Adsp could be a potential target molecule for preventing and controlling *Trichinella* transmission from pigs to humans.

## Acknowledgments

We thank Xuejin Su, Xinrui Wang and Li Yang for the technical assistance. Our thanks are also extended to express our gratitude to all the people who made this work.

## Author Contributions

**Conceptualization:** Daoxiu Xu.

**Data curation:** Daoxiu Xu, Xue Bai, Jing Xu.

**Formal analysis:** Daoxiu Xu, Xue Bai.

**Funding acquisition:** Mingyuan Liu, Xiaolei Liu.

**Investigation:** Xuelin Wang, Zijian Dong, Wenjie Shi, Yanfeng Li.

**Methodology:** Daoxiu Xu, Jing Xu, Zijian Dong, Fengyan Xu, Yanfeng Li.

**Project administration:** Daoxiu Xu, Xue Bai, Mingyuan Liu, Xiaolei Liu.

**Supervision:** Mingyuan Liu, Xiaolei Liu.

**Validation:** Jing Xu, Xuelin Wang, Xiaolei Liu.

**Writing – original draft:** Daoxiu Xu.

**Writing – review & editing:** Daoxiu Xu, Xuelin Wang, Wenjie Shi, Xiaolei Liu.

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
