## [Decision Letter · Decision Letter 0]

18 Feb 2021

Dear Mr Liu,

Thank you very much for submitting your manuscript "The immune protection induced by a serine protease from the Trichinella spiralis adult against Trichinella spiralis infection in pigs" for consideration at PLOS Neglected Tropical Diseases. As with all papers reviewed by the journal, your manuscript was reviewed by members of the editorial board and by several independent reviewers. In light of the reviews (below this email), we would like to invite the resubmission of a significantly-revised version that takes into account the reviewers' comments. 

We apologize for the delay in reviewing your manuscript. Several invited reviewers could not accept this task for several different reasons. However, all reviewers who at the end accepted reviewing your article, considered your work of importance in field of vaccines for animal diseases of veterinary importance. Yet, several major concerns on previous helminthic infections of the piglets previous to vaccination, narrow and limited analysis of cellular immune response, no measurement of specific B and T cell responses against the serine protease antigen, not suitable statistical analysis, and no mechanistic insight on the mode of action of this vaccine, among others, were raised . Thus, in its present form this article is not suitable for publication and a major revision addressing all points is required.

We cannot make any decision about publication until we have seen the revised manuscript and your response to the reviewers' comments. Your revised manuscript is also likely to be sent to reviewers for further evaluation.

Sincerely,

Hernando A del Portillo

Associate Editor

Maria Elena Bottazzi

Deputy Editor

We apologize for the delay in reviewing your manuscript. Several invited reviewers could not accept this task for several different reasons. However, all reviewers who at the end accepted reviewing your article, considered your work of importance in field of vaccines for animal diseases of veterinary importance. Yet, several major concerns on previous helminthic infections of the piglets previous to vaccination, narrow and limited analysis of cellular immune response, no measurement of specific B and T cell responses against the serine protease antigen, not suitable statistical analysis, and no mechanistic insight on the mode of action of this vaccine, among others, were raised . Thus, in its present form this article is not suitable for publication and a major revision addressing all points is required.

Reviewer's Responses to Questions

**Key Review Criteria Required for Acceptance?**

**Methods**

-Are the objectives of the study clearly articulated with a clear testable hypothesis stated?

-Is the study design appropriate to address the stated objectives?

-Is the population clearly described and appropriate for the hypothesis being tested?

-Is the sample size sufficient to ensure adequate power to address the hypothesis being tested?

-Were correct statistical analysis used to support conclusions?

-Are there concerns about ethical or regulatory requirements being met?

Reviewer #1: -Are the objectives of the study clearly articulated with a clear testable hypothesis stated?

Yes

-Is the study design appropriate to address the stated objectives?

Yes

-Is the population clearly described and appropriate for the hypothesis being tested?

No (see below)

-Is the sample size sufficient to ensure adequate power to address the hypothesis being tested? Yes

-Were correct statistical analysis used to support conclusions?

I have doubts (see below for details)

-Are there concerns about ethical or regulatory requirements being met?

No

1.- Animals:

How do the researchers check that the pigs were not previously affected with Trichinella spiralis before carrying out the immunization? Specify the type of analysis carried out.

1.- Measuring humoral immune response

Authors detailed that blood from pigs in each group was collected to evaluate the changes in specific antibodies (IgG, IgG1, IgG2a and IgM) by an indirect enzyme-linked immunosorbent assay. Authors described that titers of specific IgG and IgG subtypes were detected as described previously [Xu et al., 2020a] but this paper is focused on measuring this immune response in mice. Authors must clarify if they have used the same antibodies for mice and pigs. If they would have used the same antibodies, specify the studied carried out to check the sensitivity and specificity between species.

2.- Cytokine assay

Authors specify that serum samples of pigs were analyzed two weeks after the last immunization for cytokine determination. This sample point must be justified. Is there any previous data? Have the authors carried out a previous study to determine this sample point as suitable.

3.- Flow cytometry

The relative ratios of T cells, B cells, and neutrophils were evaluated by flow cytometry but T and B cells are not specifically measured against the antigen of interest (serine protease gene from an adult T. spiralis (Ts-Adsp)). Authors must justify why they have not specifically measured the T and B cells responding to this antigen of interest.

4.- Statistical analysis:

Reviewer has revised in detail the statistical analysis done and disagrees with some of the statistical methods used by authors:

Continuous data were analyzed with a one-way analysis of variance (ANOVA) and if this method was significant (p<0.05), no pairwise testing was specified. Referee believes that this statistical method cannot be suitable because the continuous variables analysed could not follow a normal distribution. Thus, Authors must carry out a normality test with each variable and if this variable does not follow a normal distribution (most probable event), a non-parametric test must be used instead of the one use by the authors.

Reviewer #2: The study is based on testing a candidate vaccine for T. spiralis. In that sense, the objectives are exposed in a clear manner and reflects the hypothesis behind the vaccine. The methods scheme followed is logical and allowed to complete the objectives for evaluation immune response and protection with a correct use of sample number and statistical methods. It would be better to add also an explanation of safety or secondary effects of the vaccine when administered to comply with animal welfare. 

Line 124: Preparation and identification of Recombinant Ts-Adsp:

Please explain more in detail the western blot analysis. Serum samples tested? Number of positive sera? Negative control sera? 

Line 136: In Immunization procedure and challenge: 

• Please specify vaccination dosage for group. Only the recombinant protein concentration of the dosage is explained. In the groups of PBS and PBS + Adjuvant the volumes of each component are not mentioned (proportion of antigen + adjuvant). 

In Flow cytometry analysis: 

• Line 158: Could you please specify which kind of Lysis buffer did you use? Commercially available or in-house made? In that case, could you specify the components?

• The whole process for staining cells is explained in the methods section. However, live staining is not mentioned in this section. Could you please indicate which type of live staining did you use for separate live/dead cells before gating the other markers? Could you please add the whole gating strategy?

Reviewer #3: Major comments:

1) Given the piglets were bought from a farm were they tested for infection with other helminths or treated with anti-helminthics prior to the start of the experiments? Co-infections if present could complicate the functional outcomes of the vaccination experiments.

2) The challenge experiments were only performed once. While the group sizes were reasonable given the nature of the hosts being used the vaccination and challenge experiments need to be completed several times to assess reproducibility which can vary considerably between experiments. I acknowledge the financial and technical issues associated with veterinary studies using animals like pigs do not make this easy but this is a key factor in any vaccinology study.

3) While the antigen specific serological data was very encouraging the methods used to assess cellular responses via FACs were indirect and largely unhelpful for determining anything more than a cellular response was occurring. 

Porcine PBMCs could have been collected and then stimulated ex vivo with Ts-Adsp to assess antigen specific cellular responses. This would have been particularly helpful for examining the T-cell populations and linking Th development with the serum cytokine responses presented in the manuscript.

4) One gap in knowledge that would have been useful to have addressed by the authors is to determine the specificity of antibody responses developed against Ts-Adsp. Does sera from immunized pigs (or mice) cross-react any of the other serine proteases secreted by mL1, NBL or Adult stages? Some of the T. spiralis secreted proteins are members of multi-gene families. I believe this is the case for the serine proteases that have been identified in mL1 ES. This could easily be assessed by western blot probing ES from those key stages. 

Minor comments:

1) Could the authors include additional information on the Endoxin testing kit they used to assess endotoxin contamination post-purification with the ToxOut Endotoxin removal kit.

2) Can the authors explain why the adjuvant Montanide IMS 1313 was selected for these trials rather one of the other commonly used veterinary adjuvants such as ISA71VG. 

3) The authors need to clarify what post-hoc test was used with ANOVA.

**Results**

-Does the analysis presented match the analysis plan?

-Are the results clearly and completely presented?

-Are the figures (Tables, Images) of sufficient quality for clarity?

Reviewer #1: -Does the analysis presented match the analysis plan?

Yes

-Are the results clearly and completely presented?

It can be improved (see below)

-Are the figures (Tables, Images) of sufficient quality for clarity?

It can be improved (see below)

Results must be rewritten taking into account the new possible statistical output reanalysing data (if necessary) with non-parametric test.

Line 219. Can the authors really assure that the results suggested that the cellular immune response in pigs was induced after vaccination with rTs-Adsp when no significant increase in CD8 T cells is observed?

Line 220-227. Can the authors support that the increase in B cells is only associated with the vaccination process? Is it not possible to specifically identify specific B cells reacting to Ts-Adsp?

Line 244-250. Can the author describe the results obtained in relation with larvae burden (Figure 8)? Are they statistically significant? (indicate p value). I believe that this is the case but it must be detailed in the paper.

Figure 2A and 2B is of poor quality to distinguish the three groups. Please revise the group label to make results clearer.

Reviewer #2: The results explained in the present research reflected the objectives and the original hypothesis, starting from the production and purification of the protein, to finally being tested in the animal model. The results are explained in a concise and well-organized manner.

Line 193: How many positive sera were tested? How many negative sera were tested?

Figure 4, 5 and 6. Figure quality should be improved in the X axis of bar graphics. Gating strategy should be included in the figure.

Figure 7. Figure quality should be improved.

Line 271-2: Rephrase. It is well known that the protective effect of vaccines is related to the antibody responses. (Not all vaccines enter in this category).

Reviewer #3: Major comments:

1) The protection observed in this study is reasonably similar to those observed in many other studies in rodent hosts. The limited nature of the experiments performed does not offer any clear mechanistic data for understanding the basis of vaccine efficacy. Ideally expulsion kinetics of adult worms could have been assessed however in the absence of being able to perform experiments that would require larger numbers of animal some basic assessments such as enhancement of ADCC or inhibition of larval moulting should have been tested using sera collected from vaccinated pigs. These have been done in rodent models with some success for other antigens. Even if the data is negative, if the correct controls are included it would allow the authors to narrow down the potential ways the vaccine might be reducing parasite numbers.

2) The observations presented within figure 7 are extremely problematic on a number of levels. While there is clearly a difference in the inflammation seen in control and experimental images there was no attempt to quantify this observation. Sets of histological sections would need to be assessed blindly and counting of infiltrating cells for of large number (+50) of nurse cells performed from each animal. The capsule thickness could also be measured for these nurse cells. In the absence of clear quantitative data it would not be appropriate to include this observation.

Minor comments:

1) The layer of collagen around the nurse cell is generally referred to as a capsule rather than a sac.

**Conclusions**

-Are the conclusions supported by the data presented?

-Are the limitations of analysis clearly described?

-Do the authors discuss how these data can be helpful to advance our understanding of the topic under study?

-Is public health relevance addressed?

Reviewer #1: -Are the conclusions supported by the data presented? Yes

-Are the limitations of analysis clearly described? Yes

-Do the authors discuss how these data can be helpful to advance our understanding of the topic under study?

It can be improved (see below)

-Is public health relevance addressed?

Yes

Discussion:

General comment: It could be necessary to revise it if some of the statistical output is different (see material and method section).

Authors must clearly clarify the role of cellular immunity (mainly CD8 T cells) versus humoral immunity in the outcome of Trichinella spiralis infection. It is confusing across the discussion section.

Reviewer #2: The conclusions exposed in this article complied with the data presented. However, the authors previously tested another vaccine candidate for T. spiralis in the porcine model. In that sense, there is no mention at all of this work in the discussion, neither introduction what would be interesting in terms of achieving a better immune response and protection or a worse one. Only mentioned in the methods two times. 

The present research followed almost exactly the same methodology as the previous study with a DNAse II recombinase but only including an additional experiment of cell population characterization. For that, I believe that novelty of the research is not complied and I particularly missed some more in deep comparison in discussion with other vaccines for T. spiralis tested in the porcine model (as the one already done by the group) or in another scenario, make a whole publication including the two different candidates (the one in the previous study and the one in this research) as the methodology applied is quite similar in both cases.

Reviewer #3: While the basic conclusions (ie Ts-Adsp is an ok vaccine in pigs) provided by the authors are reasonable the lack of any mechanistic data makes much of the discussion purely speculative. While it is satisfying to see the successful application of a defined Trichinella antigen in a veterinary model the scope of the study has not significantly enhanced our understanding of why this particular antigen is efficacious and its mechanisms of action of the vaccine.

**Editorial and Data Presentation Modifications?**

Reviewer #1: No comment in this section

Reviewer #2: Another Typos in the manuscript

Line 82: Abbreviation of muscle larvae should be included. 

Line 514: were detected ---- Were evaluated

Line 521: cytokines (plural) from immunized group are significantly different from

Line 523: The percentage

Line 543: Asterisks indicate that muscle larvae burden of immunized group is significantly different

Reviewer #3: Figure 7 should be removed unless quantitative data provided.

**Summary and General Comments**

Reviewer #1: I believe that this is a very interesting study and the information provided is relevant in the field of vaccinology of parasitic diseases. Unfortunately, there are no vaccines available to cope with diseases produced by helminths and any new research work is welcome in this field. The paper is well-designed, well-written but key topics must be clarified before its publication. 

I have detailed in a document the points to be addressed.

Reviewer #2: The authors previously tested another vaccine candidate for T. spiralis in the porcine model. However, there is no mention at all of this work in the discussion, neither introduction. Only mentioned in the methods two times. The present research followed almost exactly the same methodology as the previous study with a DNAse II recombinase but only including an additional experiment of cell population characterization. In that sense, I believe that novelty of the research is not complied and I particularly missed some more in deep comparison in discussion with other vaccines for T. spiralis tested in the porcine model (as the one already done by the group).

Reviewer #3: The submitted manuscript describes the outcome of T. spiralis vaccination trials in pigs using an antigen Ts-Adsp whose isolation and characterization has been published previously in a series of manuscripts from the authorship group. The authors clearly demonstrate Ts-Adsp is immunogenic in pigs and in combination with the adjuvant IMS1313 induced Th1 skewed antibody responses and mixed cellular responses in the vaccinated animals. Challenge of the vaccinated pigs demonstrated that Ts-Adsp induced a reasonable level of protective immunity as evidenced by reduced recovery of muscle stage larvae. The authors suggest but do not experimentally test the potential mechanisms underlying the protection they observe.

General Comments:

The study represents a positive progression in a veterinary model from the previously published studies on this antigen which demonstrated the potential of the Ts-Adsp as a vaccinogen in mice using different administration systems (Xu et. al. 2020 Acta Tropica and Feng et. al. 2013 J. Parasitology).

With one exception (Figure 7) the data generated is reasonably straight forward and presented clearly to the reader.

While the manuscript text written well there are several areas that could benefit from some revision or consolidation.

PLOS authors have the option to publish the peer review history of their article (what does this mean?). If published, this will include your full peer review and any attached files.

Reviewer #1: No

Reviewer #2: No

Reviewer #3: No
---

## [Editor Report · Decision Letter 1]

26 Apr 2021

Dear Mr Liu,

We are pleased to inform you that your manuscript 'The immune protection induced by a serine protease from the Trichinella spiralis adult against Trichinella spiralis infection in pigs' has been provisionally accepted for publication in PLOS Neglected Tropical Diseases.

Best regards,

Hernando A del Portillo

Associate Editor

Maria Elena Bottazzi

Deputy Editor

Queries from reviewers have been satisfactorily answered. This manuscript is therefore accepted for publication.

---

## [Editor Report · Acceptance letter]

6 May 2021

Dear Mr Liu,

We are delighted to inform you that your manuscript, "The immune protection induced by a serine protease from the Trichinella spiralis adult against Trichinella spiralis infection in pigs," has been formally accepted for publication in PLOS Neglected Tropical Diseases.

Best regards,

Shaden Kamhawi

co-Editor-in-Chief

Paul Brindley

co-Editor-in-Chief
